# Bioactive Antimicrobial Peptides as Therapeutic Agents for Infected Diabetic Foot Ulcers

**DOI:** 10.3390/biom11121894

**Published:** 2021-12-17

**Authors:** Jessica Da Silva, Ermelindo C. Leal, Eugénia Carvalho

**Affiliations:** 1Center for Neuroscience and Cell Biology, University of Coimbra, Rua Larga, 3004-504 Coimbra, Portugal; jessicasilva@cnc.uc.pt; 2PhD Programme in Experimental Biology and Biomedicine, University of Coimbra, 3004-504 Coimbra, Portugal; 3Institute of Interdisciplinary Research, University of Coimbra, Casa Costa Alemão, Rua Dom Francisco de Lemos, 3030-789 Coimbra, Portugal

**Keywords:** antimicrobial peptides, chronic non-healing wounds, diabetic foot ulcers, wound healing, bacterial and fungal infections, biofilms

## Abstract

Diabetic foot ulcer (DFU) is a devastating complication, affecting around 15% of diabetic patients and representing a leading cause of non-traumatic amputations. Notably, the risk of mixed bacterial–fungal infection is elevated and highly associated with wound necrosis and poor clinical outcomes. However, it is often underestimated in the literature. Therefore, polymicrobial infection control must be considered for effective management of DFU. It is noteworthy that antimicrobial resistance is constantly rising overtime, therefore increasing the need for new alternatives to antibiotics and antifungals. Antimicrobial peptides (AMPs) are endogenous peptides that are naturally abundant in several organisms, such as bacteria, amphibians and mammals, particularly in the skin. These molecules have shown broad-spectrum antimicrobial activity and some of them even have wound-healing activity, establishing themselves as ideal candidates for treating multi-kingdom infected wounds. Furthermore, the role of AMPs with antifungal activity in wound management is poorly described and deserves further investigation in association with antibacterial agents, such as antibiotics and AMPs with antibacterial activity, or alternatively the application of broad-spectrum antimicrobial agents that target both aerobic and anaerobic bacteria, as well as fungi. Accordingly, the aim of this review is to unravel the molecular mechanisms by which AMPs achieve their dual antimicrobial and wound-healing properties, and to discuss how these are currently being applied as promising therapies against polymicrobial-infected chronic wounds such as DFUs.

## 1. Introduction

Diabetes mellitus (DM) is a chronic disease, with a continuously increasing worldwide prevalence, that affected 463 million adults globally, in 2019 [1,2,3]. In Europe alone, the DM prevalence was about 59 million adults in 2019, and it is estimated to rise to over 68 million by 2045, representing an increase of 15% [1,3]. Similarly, the DM-associated complications are also expected to increase [1,2,3,4]. Indeed, diabetic foot and lower limb complications affect between 40 to 60 million people globally, representing an important source of morbidity in people with DM [1,3]. About 15% of patients with DM will develop foot ulcers in their life time, requiring prolonged hospitalizations and amputations in 85% of the cases [1,2,3,4,5,6].

A diabetic foot ulcer (DFU) is a devastating and costly complication of diabetes, consisting of deep tissue lesions associated with both peripheral neuropathy and peripheral vascular disease [7,8]. DFU represents a severe public health problem with an urgent need for new effective treatments, which are crucial to reduce the associated high morbidity and mortality rates, as well as to reduce the economic and social burden [2,6,8]. The persistent hyperglycemia, chronic inflammation, hypoxia, peripheral neuropathy, impaired angiogenesis, and difficulty to fight infections in diabetes are factors that impair the wound healing progress. Importantly, around 60% of DFUs become infected, predominantly with bacterial colonies of *S. aureus* and *C. striatum*, and fungal colonies of *C. albicans* [9,10,11,12,13,14,15,16]. Moreover, anaerobic bacteria of the Bacteroidales order, namely *Bacteroides* spp. and *Prevotella* spp., have also associated with non-healing DFUs, whereas fungal pathogens have been highly associated with wound necrosis and poor clinical outcomes [17,18,19,20]. Still, the literature is almost exclusively focused on Gram-positive and Gram-negative bacteria, and few studies have considered the prevalence of anaerobic bacteria and fungi in DFUs. Therefore, multi-kingdom microbiome infection control is imperative for the management of this kind of infected wounds. Indeed, wound microbiota and microbial biofilms are thought to contribute to the failure of chronic wound to heal; hence, the control of pathogen infection is a good therapeutic solution, since it can improve the microenvironment and promote sustained healing over time.

Antimicrobial peptides (AMPs) are endogenous peptides found in different organisms, particularly in the skin, that act as a first line of defense against infection [21,22]. Furthermore, these molecules not only play key roles in fighting infection through broad-spectrum antimicrobial activity against Gram-positive and Gram-negative bacteria, viruses, and fungi, but they also play important roles in wound healing [23,24,25,26]. Nonetheless, endogenous AMP expression and/or activity can be dysfunctional under certain conditions, such as diabetes, making them not able to promote adequate healing and fight infection at the wound site. Accordingly, suitable therapeutic strategies for the management of polymicrobial-infected wounds should rely on the performance of chemical modifications and/or the use of novel delivery systems for exogenous AMPs, to increase their stability, reduce their toxicity, enhance their dual antimicrobial and wound-healing activities, and improve their targeting and prolonged delivery at the wound site.

Therefore, this review aims to describe the molecular mechanisms by which AMPs achieve their dual multi-kingdom antimicrobial and wound-healing properties. In addition, it will describe how these AMPs are currently being applied as promising therapies to combat polymicrobial infection in DFUs.

## 2. Diabetic Foot Infection

Wound healing comprises a complex and dynamic series of cellular and biochemical events which consists of the following four overlapping phases: hemostasis, inflammation, proliferation, and remodeling [5,27,28,29]. The hemostasis phase begins with constriction of the injured blood vessels and activation of platelets to form a fibrin clot to stop the bleeding [27,30]. Subsequently, the inflammatory phase initiates with the recruitment of neutrophils to the clot as a first line of defense against pathogens to remove debris, in order to provide a propitious environment for wound healing [27,28,30]. Neutrophils reach their peak population between 24 and 48 h after injury, after which they reduce greatly in number, and macrophages, in turn, arrive at the wound site and continue clearing debris [27,28]. Macrophages secrete growth factors and proteins that attract adaptive immune system cells to the wound site, such as Langerhans cells, dermal dendritic cells and T cells, which are involved either in the clearance of cellular debris or in the combat of infection [27,28]. Once the wound has been cleaned out and the inflammation decrease, the proliferative phase occurs with the following three different stages: filling of the wound with granulation tissue, contraction of the wound margins, and covering of the wound with epithelial cells, also called re-epithelization [27,28,30]. Finally, the remodeling phase takes place with collagen fiber reorganization, tissue remodeling and maturation, and an overall increase in the tensile strength can be observed [27,28,30].

However, besides its complexity, the healing process is also susceptible to interruption or delay, due to impairment of local and systemic factors that are important in the healing process. Chronic non-healing wounds often develop in people with diabetes. If wounds do not heal within 12 weeks, they are defined as chronic wounds according to the Food and Drug Administration [29,31,32]. Furthermore, in the presence of conditions such as hyperglycemia, chronic inflammation, hypoxia, peripheral neuropathy, impaired angiogenesis, and infection the wound healing progress in diabetes can become stalled [31,32].

Diabetic foot infections (DFIs) are defined by invasion and multiplication of microorganisms in diabetic non-healing wounds and are associated with tissue destruction and/or alterations in the host’s inflammatory response [9,33,34]. DFIs are among the most serious and frequent complications in people with diabetes. They are estimated to develop in about 60% of all DFU cases and represent an important source of morbidity in these patients [9,35,36,37]. Several aspects of the wound microbiology influence the development of DFI, including the microbial load, the microbe diversity, the existence of pathogenic microorganisms, and the synergistic association amongst microbial species [9,33,35,38]. Among the most predominantly identified bacteria in DFUs are not only Gram-positive bacteria, such as *S. aureus* (MSSA—methicillin-susceptible *Staphylococcus aureus*, and MRSA—methicillin-resistant *Staphylococcus aureus*), *Streptococcus β-hemolytic* and *C. striatum*, but also Gram-negative bacteria, such as *P. aeruginosa*, *E. coli*, *A. baumannii*, *Proteus* spp., *Enterobacter* spp., and *Citrobacter* spp., in addition to some anaerobes deeper in the wound bed, such as *Bacteroides* spp., *Prevotella* spp., *Clostridium* spp., and *Peptostreptococcus* spp. (Table 1).

Furthermore, DFUs have a polymicrobial basis, and the risk for the diabetic foot syndrome development is mostly associated with mycotic infections [11,12,13,14,33]. However, few studies have considered the prevalence of fungal colonies in DFUs. Indeed, more than a quarter of DFUs undergo fungal infection, but remain undetected or undiagnosed by regular and standard microbiology laboratory protocols in the DFU clinics, in most cases, as it also happens with anaerobic bacteria [14,20,39,40]. It has also been demonstrated that patients with higher systemic glycosylated hemoglobin levels, such as diabetic patients, have significantly more fungal infections, which contribute to delayed wound healing [14]. Importantly, the mycobiome represents a scaffold for bacterial attachment and provides additional protection from external threats, promoting the formation of multi-kingdom biofilms [19,20]. Moreover, increased fungal pathogens in DFUs have been highly associated with wound necrosis and poor clinical outcomes [19,20]. The fungi most commonly isolated are *Candida* spp., *Trichophyton* spp., *Aspergillus* spp., *Trichosporon* spp., and *Cladosporium herbarum* (Table 1).

The formation of microbial biofilm in DFUs, defined as a structured arrangement of microorganisms in a self-produced polysaccharide matrix with transformed phenotype and growth patterns, has been related to wound chronicity and infection [9,35,38,41]. Biofilms may be explained by the organization of these microorganisms into functionally equivalent pathogroups (FEP) in DFUs, where pathogenic and commensal microorganisms co-aggregate symbiotically in a pathogenic biofilm for more efficient nutrient cycling and enhanced protection from external threats, further promoting chronic infection [17,35,42,43]. Additionally, it is noteworthy that biofilm-forming microbial colonies are 10 to 1000 times more resistant to antimicrobials, including both antibiotics and antiseptics, in comparison with planktonic ones, which consists of free-floating microorganisms. Therefore, it is urgent to find effective treatments for chronic infected DFUs with a polymicrobial basis. The combination of multidisciplinary treatment approaches should help to overcome some of the DFI-related hurdles [17,35,38]. As a result, the role of AMPs with antifungal activity in wound management needs to be considered and further investigated, in association with antibacterial agents, such as antibiotics and AMPs with antibacterial activity, or alternatively the application of a broad-spectrum antimicrobial agent that targets both bacteria and fungi.

## 3. Antimicrobial Peptides

Antimicrobial peptides (AMPs), also known as endogenous host defense peptides, are naturally abundant peptides found in bacteria, plants, insects, amphibians, reptiles, and mammals. These peptides play essential roles in the innate immune response and contribute to the first line of defense against infection [21,22,26,45]. Upon injury and infection, the innate immune system is activated and leads to the production of these small molecules by different resident cells of the skin such as keratinocytes, the predominant cell type of the epidermis [26,41,42,43]. Indeed, pathogen-associated molecular patterns (PAMPs), such as lipoarabinomannan, lipopolysaccharides and proinflammatory cytokines, are recognized by the innate immune system, leading to the up-regulation and overexpression of AMPs to promote a fast and effective response to injury and infection [23,29,45,46].

AMPs are composed of 15 to 50 amino acids, are generally positively charged, form amphipathic structures, and are classified into different categories according to their primary structures and topologies, including human endogenous β-defensins (hBDs) 1–3, cathelicidin antimicrobial peptide (LL-37) and dermcidins [41,45,46,47,48,49]. The two most predominant types of AMPs in human skin include hBDs and cathelicidins, particularly hBDs 1–3 and LL-37, with their primary, secondary and tertiary structures, and their related physicochemical properties presented in Table 2 [30,46,49,50]. These physicochemical properties, including length, molecular weight (MW), isoelectric point (pI), net charge, and hydrophobicity, are important to predict their antimicrobial potential for further clinical application.

The hBDs 1–3 and LL-37 have a peptide length below 50 amino acids (aa) and a relatively similar MW, thereby being referred to as AMPs, but also as small peptides (Table 2). The small length and low MW of these peptides can promote their insertion of the peptide into the microbial membrane, contributing to their higher antimicrobial activity [51]. Moreover, all four of these endogenous AMPs exhibit a high positive net charge, ranging from +4 to +11, and a relatively similar isoelectric point, ranging from 8.55 to 11.15 (Table 2). This net positive charge is a requirement for their antimicrobial potential, in order to permeabilize the negatively charged membranes of microbes [22,46,47,52]. In addition, the hydrophobicity properties are also crucial for partial or total insertion of AMPs into the membrane’s hydrophobic core [51]. This AMP membrane insertion will enable the destabilization of the bilayer and/or promote the cell depolarization, denoting the importance of a high AMP hydrophobicity for antimicrobial potential [51]. All four peptides (hBDs 1–3 and LL-37) exhibit a high hydrophobicity value ranging from +28.98 to +45.26 kcal/mol, another important property highlighting their antimicrobial potential (Table 2). Furthermore, the secondary and tertiary structure properties are another key feature influencing the biological function of these small peptides [51]. Regarding the secondary structure, hBDs 1–3 present a mixed α-helix + β-strand conformation, whereas LL-37 exhibits only an α-helix arrangement (Table 2) [30,49], which are the most common conformations in AMPs [53]. In regard to their tertiary structure, hBDs 1–3 present a relatively similar “defensin-like” topology, i.e., a core consisting of three antiparallel β-sheets interconnected with three intramolecular disulfide bridges flanked by an α-helix segment, all together stabilized by a disulfide bridge, making them members of the defensin family. On the other hand, LL-37 presents a predominant α-helical conformation, making it the only human member of the cathelicidin family (Table 2) [30,52,54]. As a result, these physicochemical properties greatly influence the activity and the potential of AMPs, therefore highlighting the need for the inclusion of such parameters when evaluating AMPs and selecting them for further clinical application.

Besides their well-known broad-spectrum antimicrobial activity against Gram-positive and Gram-negative bacteria, viruses, and fungi, some AMPs also play key roles in wound healing by promoting cell migration and proliferation, angiogenesis, chemokine and cytokine production, and wound closure (Figure 1) [23,24,25,26,50]. Therefore, these aforementioned AMPs are usually referred to as peptides with dual antimicrobial and wound-healing properties [46].

AMPs can achieve direct eradication of microbes by disrupting microbial membranes through pore formation and by interacting with intracellular targets, such as hBD-2, hBD-3 and LL-37 (Figure 1) [22,47,49]. This antimicrobial mechanism of action by the disruption of microbial membranes is based on the permeabilization of negatively charged membranes of the microbes followed by microbial lysis, due to the positive charge of AMPs [22,46,47]. On the other hand, AMPs can also modulate the host immune system by the recruitment and activation of immune cells through the induction of chemokine and cytokine production, and, therefore, enhancing indirect pathogen killing and clearance and controlling inflammation, namely hBD-2, hBD-3, LL-37 and dermcidin-1L (Figure 1) [25,30,47,49].

Importantly, some AMPs are also able to promote re-epithelization and wound closure through activation of receptor-signaling mechanisms responsible for cell proliferation and migration, such as hBD-2, hBD-3 and LL-37 (Figure 1) [54]. In addition, they can also support angiogenesis by the induction of endothelial cell tube formation and up-regulation of angiogenic proteins, namely LL-37 (Figure 1) [54]. Furthermore, they can enhance extracellular matrix synthesis, promote the contraction capacity of fibroblasts by inducing fibroblast-to-myofibroblast differentiation, and enhance wound healing by increasing α-smooth muscle actin expression by fibroblasts (Figure 1) [23,47,49,55].

These small peptides, both from natural sources or synthetically produced, have been investigated in depth in the last years, and some of them are even in clinical trials. Indeed, LL-37 (ropocamptide) from Promore Pharma AB is currently under phase IIb for treating venous leg ulcers [55], and pexiganan (Locilex^®^) from Dipexium Pharmaceuticals, Inc., an analogue of peptide magainin II from frog skin of *Xenopus laevis*, was under phase III for treating mild infected DFUs [52]. Unfortunately, Locilex^®^ was discontinued for not meeting greater primary or secondary clinical endpoints versus the vehicle plus standard of care [52]. Moreover, FirstString Research, Inc. also developed a Granexin^®^ Gel that contains the synthetic aCT1 peptide for treating DFUs [56]. This formulation was under phase III until May 2020, and has since been terminated without safety or efficacy concerns, with final data to be published [56].

Therefore, AMPs need to be further investigated as promising alternatives to conventional antibiotics to overcome the emergence of multidrug-resistant (MDR) microorganisms and as an attractive strategy for polymicrobial-infected DFUs, due to their dual antimicrobial and wound-healing properties [21,25,47,57].

## 4. Changes of Endogenous AMPs in DFUs

Wound healing and infection control are efficiently carried out in the skin by AMPs and other molecules, such as growth factors. Important endogenous AMPs participating in these events include hBDs, LL-37, and dermcidins, which are naturally abundant in different organisms, particularly in the skin [23,53,58]. However, their expression levels and/or activity may be altered under certain conditions, including diabetes, leading to inadequate infection control, and contributing to impaired wound healing.

Lan et al. have shown that when human keratinocytes isolated from normal adult foreskin are cultured in vitro under a high glucose environment for 7 days, hBD-2 expression is reduced through the downregulation of signal transducer and activator of transcription 1 (STAT-1) signaling [59]. Indeed, STAT-1 is a transcription factor that is involved in the upregulation of many genes, due to a signal by either type I, II or III interferons, suggesting that functional STAT-1 signaling is required to achieve optimal hBD-2 transcription. In addition, the skin of streptozotocin (STZ)-induced diabetic rats showed inadequate β-defensin expression after wounding compared with skin from control rats, contributing to poor diabetic wound healing [59]. Moreover, Gonzalez-Curiel et al. have determined that patients with type 2 diabetes express lower levels of CAMP (LL-37) and DEFB4 (hBD-2) genes in peripheral blood cells, which could explain the higher susceptibility to infectious diseases [23]. Moreover, Galkowska et al. have revealed that chronic wounds, grade 2–4 DFUs according to the Wagner’s classification and venous calf ulcers, present underexpression of hBD-2 in comparison to normal skin, which may point to the involvement of this peptide in the chronicity of ulcers [60]. Conversely, Rivas-Santiago et al. have demonstrated that hBDs were overexpressed in biopsies from grade 3 DFUs according to the Wagner’s classification, whereas LL-37 is under expressed or absent in comparison with biopsies from healthy skin donors [50]. Although Rivas-Santiago et al. found that hBDs are expressed in DFUs, their activity seems to be inefficient to fight infection and promote proper wound healing [50].

All together, these results suggest that though some endogenous AMPs are expressed in DFU, their expression level and activity is not appropriate, highlighting the need to restore the expression level and enhance the activity of these peptides at the wound site. When doing this, one needs to bear in mind factors that weaken their function, such as those found in the diabetic microenvironment, protease degradation and serum inactivation.

Nonetheless, neither the increase of PAMPs to induce up-regulation and overexpression of AMPs nor the increase of AMPs itself should be used as therapeutic approaches, due to undesirable side effects in patients that PAMPs may induce and due to the potential toxicity of free AMPs, respectively [23,48]. Therefore, a suitable strategy may rely on the performance of chemical modifications and/or the use of delivery systems, in order to increase the stability of these peptides in the DFU microenvironment, reduce their toxicity, enhance their dual antimicrobial and wound-healing activities, and improve their targeting and prolonged delivery, including deeper in the wound bed [41,48].

## 5. Endogenous and Synthetic AMPs as Promising Therapeutic Agents for Infected Wounds

Several AMPs are being studied as promising therapies to combat infected non-healing wounds, and some of them are even in clinical trials as formerly referred, either free or loaded onto a delivery system. Table 3 and Table 4 summarize the AMPs being explored as promising therapies for chronic non-healing wounds and their respective roles in antimicrobial and wound-healing activities, either free or loaded on a delivery system, respectively. All AMPs are presented according to the following criteria: (1) free vs. chemically modified or loaded on a delivery system vs. chemically modified and loaded on a delivery system; (2) endogenous vs. synthetic; and (3) alphabetic order.

### 5.1. Free AMPs

In recent years, bare AMPs have been broadly investigated to unravel their dual antimicrobial and wound-healing properties to be used as therapeutic approaches for infected chronic wounds (Table 3). One example is the study by Gonzalez-Curiel et al., who have demonstrated *in vitro*, that free endogenous hBD-2, a 41-amino acid peptide, has antimicrobial activity against *E. coli* and promotes keratinocyte migration in primary cultures developed from skin biopsies of patients with DFUs, at a concentration of 0.8 µg/mL [23]. Similar results were also obtained with LL-37, a peptide consisting of 37 amino acids, at a concentration of 0.1 µg/mL [23]. In addition, Liu et al. have studied CW49, a short peptide with an 11-amino acid sequence, identified in frog skin of *Odorrana graham*. This peptide was shown to have strong angiogenic ability and a significant anti-inflammatory effect, but it had little effect on re-epithelialization, being applied twice a day for 12 days at a concentration of 200 µg/mL in full-thickness dermal wounds, in both normal and diabetic animals [61]. Moreover, Steinstraesser et al. demonstrated the lack of cytotoxicity towards immortalized human keratinocyte HaCaT cells and primary human fibroblasts of synthetic innate defense regulator-1018 (IDR-1018), a 12-amino acid peptide, compared to LL-37, at a concentration of up to 200 µg/mL (*p* < 0.0001) [62]. IDR-1018 also exhibited enhanced wound-healing and antimicrobial potential in *S. aureus*-infected porcine wounds, as well as in non-diabetic murine wounds at a concentration of 200 µg/mL, applied every 48 h for 10 or 14 days [62]. However, this wound-healing effect was not demonstrated in the diabetic murine wounds, suggesting that one or more signaling pathways by which IDR-1018 acts may be impaired in the diabetic animals [62]. Additionally, Marin-Luevano et al. have also unraveled the properties of IDR-1018 in a high glucose environment [57]. This peptide was shown to have angiogenic ability and anti-inflammatory effect at both concentrations of 25 or 50 µg/mL, as well as promote the migration of endothelial cells under conditions of hypoxia at a concentration of 25 µg/mL, while epithelial cells migration increased only under conditions of normoxia [57]. Furthermore, pexiganan is another peptide, with a 22-amino acid sequence that has been studied, particularly for its broad-spectrum antimicrobial activity. According to Flamm et al., pexiganan has demonstrated antimicrobial activity against several Gram-positive and Gram-negative bacteria, including *E. coli*, *E. cloacae*, *Citrobacter species*, *P. vulgaris*, *M. morganii*, *K. pneumoniae*, *S. marcescens*, *P. aeruginosa*, *A. baumannii* (resistant to ≥4 antimicrobials), *S. agalactiae*, *S. pyogenes*, *E. faecium*, and even MSSA and MRSA, with an overall minimum inhibitory concentration (MIC) of MIC_50_ = 16 µg/mL, excepted for *M. morganii*, *P. vulgaris*, and *S. marcescens* that presented MIC_50_ > 512 µg/mL [58].

Despite several studies unravelling the properties of these promising peptides to be used as therapeutic approaches for chronic infected wounds, as alternatives to antibiotics, free AMPs still present limited activity due to their susceptibility to the microenvironment found in non-healing wounds, and their inherent toxicity [41,63]. A way to overcome some of these hurdles has been to perform chemical modifications of the peptides (Table 3). Among the main chemical modifications that have been investigated and have shown specific advantages to improve specific AMP characteristics are: lipidation (covalent binding of a lipid group to a peptide); glycosylation (covalent attachment of a glycan, or also termed carbohydrate, to a peptide); guanidination (conversion of some or all of the lysine residues to homoarginine residues); hydrazidation (binding of a hydrazide to a peptide); and small molecule conjugation (incorporation of a small molecule in AMPs, such as antibiotics, ionic liquids, or even small peptides, among others) [64,65]. Lipidation and glycosylation have mainly been explored as chemical modifications for solubility and activity improvement and stability enhancement against protease degradation, whereas the other chemical modifications have also been applied for antimicrobial activity enhancement [64,65]. Gomes et al. have synthesized 3.1-PP4, a 16-amino acid peptide, by developing hybrid peptide constructs combining the wide spectrum antimicrobial peptide 3.1 and the collagen-inducing peptide PP4, one important example of small molecule conjugation of two peptides [66]. This hybrid peptide had low toxicity towards HFF-1 human fibroblasts (half maximal inhibitory concentration—IC_50_ = 134 ± 10 µg/mL) and antimicrobial potential against *E. coli* (MIC = 2 µg/mL), *P. aeruginosa* (MIC = 4.2 µg/mL), and even against MDR isolates of *K. pneumoniae*, *E. coli*, and *P. aeruginosa* (1 < MIC < 8.2 µg/mL) [66]. The peptide 3.1-PP4 also interfered with the formation of *K. pneumoniae* biofilms of resistant clinical isolates [66]. More recently, they have also synthesized its N-methyl imidazole derivative MeIm-3.1-PP4, a chemical modification that improved its solubility and enzymatic stability towards tyrosinases. In addition, PP4-3.1 and its N-methyl imidazole derivative MeIm-PP4-3.1 were also synthesized via another chemical modification that could broaden its spectrum activity, including against fungal pathogens, relative to its reversed isomer 3.1-PP4 [67]. Interestingly, PP4-3.1 showed the highest activity against Gram-positive and Gram-negative bacteria, including MDR isolates (0.8 ≤ MIC ≤ 5.7 µM), either in planktonic or biofilm form, as well as against relevant *Candida* spp. [67]. It is noteworthy that MeIm-PP4-3.1 was almost twice more cytotoxic than PP4-3.1 in HaCaT cells (IC_50_ = 5.7 ± 1.0 or 13.0 ± 1.0 µM, respectively), highlighting the higher potential of PP4-3.1 [67]. Furthermore, Mi et al. have produced A-hBD-2, a 41-amino acid peptide, through the replacement of the GIGDP unit on the N-terminal of hBD-2 by APKAM [26]. This modification improved the structural stability of hBD-2 and led to no cytotoxicity in HaCaT cells, at a concentration of up to 100 µg/mL after 24–72 h of incubation, while also improving its antimicrobial activity against *S. aureus*, at concentrations of 50, 70, and 100 µg/mL, when compared to hBD-2 [26]. Moreover, these authors demonstrated the potential of A-hBD-2 to promote migration and proliferation of keratinocytes at a concentration of 20 µg/mL, via the phosphorylation of epidermal growth factor receptor (EGFR) and the signal transducer and activator of transcription 3 (STAT3). In addition, A-hBD-2 (20 µg/mL) decreased the terminal differentiation of keratinocytes, enhanced the mobilization of intracellular calcium ions (Ca^2+^), and promoted wound healing of full-thickness wounds in a rat model, suggesting that A-hBD-2 may be a promising candidate therapy for chronic wounds [26]. Similarly, Mouritzen et al. have assessed the potential of bovine lactoferricin (LFcinB), a 25-amino acid peptide derived from acidic hydrolysis of bovine lactoferrin (bLf), in diabetic wound healing [68]. LFcinB promoted keratinocyte migration in vitro and ex vivo at a concentration of 25 µg/mL, and enhanced wound healing in a type 1 diabetic mouse model at both concentrations of 12.5 or 25 µg/wound, applied topically over 10 consecutive days (twice the first 2 days and then once daily) [68]. Moreover, LFcinB had antimicrobial activity against *B. pumilus* and *S. aureus*, and increased *S. xylosus* prevalence, a commensal bacterium of the skin, in the type 1 diabetic mouse model [68]. It was also shown to induce angiogenesis and collagen deposition, while decreasing oxidative stress and the M1/M2 macrophage ratio, suggesting a reduction of inflammation in wounds of this diabetic mouse model [68]. In turn, Kim et al. have synthesized SHAP1, a 19-amino acid peptide, through the addition of the two capping motifs APKAM and LQKKGI into the N- and C-terminal ends, respectively, to ensure structural stability of the secondary structure of the entire peptide, irrespective of surrounding salt concentration [69]. They also showed that this peptide had no cytotoxicity towards human erythrocytes and HaCaT cells up to a concentration of 200 µM, and it proved to have greater stability to protease exposure in the wound fluid, such as human neutrophil elastase and *S. aureus* V8 proteinase [69]. Moreover, they revealed that the SHAP1 peptide had stronger wound closure activity compared to LL-37 in vitro by inducing HaCaT cell migration, at a concentration of 1 µM, and accelerated healing of full-thickness excisional wounds in mice at a concentration of 1 µM/wound, applied one time a day for 2 days post-injury. In addition, it had potent antimicrobial activity against *S. aureus*, and enhanced wound healing in *S. aureus*-infected murine wounds at a concentration of 1 µM/wound, applied one time a day for 2 days post-injury [69]. Finally, Tomioka et al. have developed a synthetic stable and short peptide, SR-0379, with 20 amino acids, including a lysine residue that has been converted to D-lysine to improve its stability [47]. This peptide (10 µg/mL) promoted the proliferation of normal human dermal fibroblast cells via the PI3 kinase-Akt-mTOR pathway through integrin-mediated interactions. SR-0379 also revealed antimicrobial activity not only against bacteria (*E. coli*, *P. aeruginosa*, and *S. aureus*), including drug-resistant bacteria (MRSA and *A. baumannii* MDR), but also against fungi (*C. krusei*, *T. mentagrophytes*, and *T. rubrum)* [47]. Additionally, they also demonstrated that SR-0379 induced wound healing in vivo in the following two different wound-healing models in rats: full-thickness wounds under a diabetic conditions (at a concentration of 200 µg/mL, applied at each time point—days 0, 6, 13 and 20) and acutely infected full-thickness wounds with *S. aureus* (at a concentration of 1000 µg/mL, applied at each time point—days 8 and 15) [47].

### 5.2. Loaded AMPs on Delivery Systems

Nonetheless, chemical modifications may not be enough to fully improve the properties of bare AMPs. Therefore, other appropriate strategies have been made available, namely the encapsulation in delivery systems. This will not only increase the stability of these peptides in the DFU microenvironment, but it may also reduce their inherent toxicity, and enhance their dual antimicrobial and wound-healing capacity [41,48,63]. In addition, it will improve their targeting and prolonged their delivery, optimizing their effectiveness for treating non-healing infected wounds [41,48,63]. Different approaches have been developed in this way (Table 4). Bolatchiev et al. have developed a niosomal gel made of silicon to encapsulate separately the human neutrophil peptide-1 (HNP-1, or α-defensin-1), a 75-amino acid peptide, or the human β-defensin-1 (hBD-1), a 47-amino acid peptide. They have demonstrated their antimicrobial activity against MRSA-infected wounds in rats, and against MSSA and MRSA isolated from patients with DFIs, at peptide concentrations of 2 or 1 µg/mL, respectively (MIC = 1 µg/mL for MSSA and MIC = 0.5 µg/mL for MRSA) [70]. Importantly, these authors have demonstrated that there was no in vitro synergistic action based on the calculated fractional inhibitory concentration index of these two peptides in combination with cefotaxime, a third-generation broad-spectrum bactericidal cephalosporin antibiotic, against MSSA, as well as against MRSA, since this strain of staphylococci has natural resistance to cefotaxime [70]. Another delivery system that has been tested by Santos et al. is a guar gum gel used for the topical delivery of nisin, a 34-amino acid peptide that belongs to class I bacteriocins and is produced by the bacteria *Lactococcus lactis* [25,71]. In 2016, this guar gum gel, used as a nisin delivery system, was shown to exhibit antimicrobial activity against *S. aureus* biofilm-producing isolates collected from DFU patients, including multidrug-resistant clinical isolates (overall MIC = 180.8 53.9 µg/mL) [25]. A few years later, it was shown to have an even higher inhibitory efficacy against *S. aureus* biofilm formation from DFI patients, when combined with chlorhexidine, an antiseptic agent, at a peptide concentration of 22.5 µg/mL and at an antiseptic agent concentration of 6 µg/mL, suggesting a synergistic action between the peptide and the antiseptic agent, contrary to Bolatchiev et al. [71]. However, no significant differences were shown between the efficacy of this combination and the conventional antibiotic-based protocols regarding biofilm eradication [71]. Furthermore, Grek et al. have developed a hydroxyethyl cellulose gel, called Granexin^®^, for the topical delivery of aCT1, a 25-amino acid synthetic peptide mimetic of the C-terminus of connexin43 (Cx43), which is known to have roles in dermal wound healing [56]. This formulation, applied topically with a peptide concentration of 100 µM on days 0 and 3, and then weekly from weeks 1–12, was shown to decrease ulcer areas, to promote ulcer re-epithelialization, and to decrease time-to-complete-ulcer closure in DFU patients within a randomized, investigator-blinded, multi-center clinical trial [56]. It is noteworthy that aCT1-containing Granexin^®^ was under a phase III clinical trial until May 2020, and has since been terminated without safety or efficacy concerns, with final data to be published [56].

Similarly, Bayramov et al. have assessed the efficacy of the following three different types of formulations: a gel containing 1.5% peptide, a Stratex^®^ dressing coated with ASP-1 gel to obtain 0.74 mg/cm^2^ peptide, and a hydrophilic polyurethane (PU)-based dressing containing 0.66 mg/cm^2^ peptide, for the topical delivery of the following two synthetic peptides separately: ASP-1, a 24-amino acid peptide and ASP-2, a 12-amino acid peptide [72]. All the three delivery systems loaded either with ASP-1 or ASP-2 induced in vitro eradication of mono- and polymicrobial biofilms of MDR pathogens, including *S. aureus*, *A. baumannii*, *K. pneumoniae*, *P. aeruginosa*, and MRSA, and presented a higher biocompatibility index (BI) when compared to free ASP-1 or free ASP-2, with a more favorable BI for ASP-2, in primary human epidermal fibroblast cells [72]. In turn, Zhao et al. have developed a glucose oxidase (GOx)-loaded hydrogel formed by the self-assembly of an heptapeptide, IKYLSVN, known for its antimicrobial properties, with a peptide concentration of 10 mg/mL [73]. This formulation was shown to have antimicrobial activity against *S. aureus* in vitro cultures and to reduce blood glucose concentration of diabetic patients [73]. Additionally, Comune et al. have developed a gold-nanoscale formulation (gold nanoparticles—Au NPs) to carry synthetic LL-37, a modified form of the endogenous LL-37 with a C-terminal cysteine [48]. This gold-nanoscale LL-37 delivery system, with a peptide concentration of about 2.34 µg/mL, increased the phosphorylation of EGFR and extracellular signal-regulated protein kinase 1/2 (ERK1/2), and it promoted the migratory properties of keratinocytes in vitro [48]. They also observed higher wound-healing activity and higher expression of collagen, interleukin 6 (IL6), and vascular endothelial growth factor (VEGF) after intradermal administration of LL-37-Au NPs at several sites around the wound, when compared to free LL-37 or empty Au NPs in an in vivo mouse model of full-thickness excisional wounds [48]. Furthermore, Lipsky et al. have produced a cream for topical delivery of pexiganan and tested its efficacy in a randomized, controlled, double-blinded, multicenter clinical trial for treating diabetic patients with a mildly infected DFU, when compared to oral ofloxacin antibiotic [74]. These authors demonstrated similar results between pexiganan 0.8% topical cream and oral ofloxacin treatments applied twice daily for 14 days along with standard local wound care, regarding clinical improvement, overall microbiological eradication against *S. aureus*, *E. coli*, *E. cloacae* and *S. marcescens*, *P. aeruginosa*, *Enterococcus species*, MSSA and MRSA, and wound healing improvement [74]. However, bacterial resistance to ofloxacin emerged in some of the patients, but not against pexiganan, suggesting that topical pexiganan could still be an encouraging alternative to oral antibiotic therapy in treating patients with mildly infected DFU [74].

By combining chemical modifications and the use of delivery systems, Song et al. used the synthetic Cys-KR12 peptide, originated from LL-37, and immobilized it onto a silk-fibroin (SF) nanofiber membrane, with peptide concentrations of 50, 100, 200, and 500 µg/mL [75]. This Cys-KR12-immobilized SF nanofiber membrane, containing 200 or 500 µg/mL of peptide, exhibited antimicrobial activity against the following four pathogenic bacterial strains: *S. aureus*, *S. epidermidis*, *E. coli* and *P. aeruginosa* without biofilm formation [75]. Moreover, these authors have demonstrated that this system promoted proliferation of keratinocytes and fibroblasts in vitro, enhanced differentiation of keratinocytes, and repressed lipopolysaccharides (LPS)-induced tumor necrosis factor alpha (TNF-α) expression of monocytes [75]. Furthermore, Sultan et al. have developed a novel peptide-based bioadhesive hydrogel formulation [76]. They synthesized K11R-K17R peptide, a 24-amino acid peptide, derivative of Histatin-5 (Hst-5), through substitution of lysine residues at positions 11 and 17 with arginine residues, to produce a stable variant peptide that is resistant to proteolytic degradation [76]. In addition, this peptide was used with the FDA-approved hydroxypropyl methylcellulose (HPMC)-based bioadhesive hydrogel as a delivery system to evaluated its efficacy in vitro [76]. This formulation, with a peptide concentration of 2 mg/mL, was shown to have antimicrobial activity against fungal *C. albicans* strains, resistant to traditional antifungals, in addition to promoting cell proliferation and migration of human oral keratinocytes [76].

Moreover, Gawande et al. have formulated a wound gel combining DispersinB^®^, an antibiofilm enzyme, with Pluronic F-127, a gelling agent, and KSL-W, a synthetic 10-amino acid peptide, representing an analogue of KSL that is known to have antimicrobial activity [77]. This formulation, with a peptide concentration of 125 or 250 µg/mL, had in vitro antibiofilm and antimicrobial activity against chronic wound infection associated biofilm-embedded bacteria, including MRSA (250 µg/mL), *S. epidermidis* (250 µg/mL), Coagulase-negative *Staphylococci* (CoNS) (250 µg/mL), and *A. baumannii* (125 µg/mL) [77]. Furthermore, Riool et al. have developed an HPMC gel to deliver TC19, a synthetic 15-amino acid peptide, derived from the human thrombocidin (TC)-1-derived peptide L3 [78]. This peptide had reduced cytotoxicity in normal human dermal fibroblasts, at a peptide concentration of up to 80 µM, after 1 h and 4 h of incubation. It also had efficient and rapid antimicrobial activity against several bacterial species of the ESKAPE panel, including *E. faecium* (MDR), MRSA, *K. pneumoniae* (MDR), *A. baumannii* (MDR), *P. aeruginosa* (PDR—pandrug-resistant, meaning non-susceptibility to all agents in all antimicrobial categories) and *E. cloacae* (MDR) [78]. Additionally, TC19 reduced bacterial resistance in vitro, and reduced pro-inflammatory activity of bacterial cell envelope components [78]. Then, the 2% TC19-containing HPMC gel, was tested and it was shown to increase antimicrobial activity against MRSA and *A. baumannii* (MDR) in a murine superficial wound infection model 4 h after its topical application [78]. Finally, Lin et al. have produced an alginate/hyaluronic acid/collagen (Alg/HA/Cil) wound dressing to immobilize Tet213, a synthetic 10-amino acid peptide, consisting on the cysteinylated form of HHC36 peptide, known to have high antimicrobial activity [79]. This Tet213-loaded dressing, with a peptide concentration of 500 µg/mL, was shown to exhibit in vitro antimicrobial activity against *E. coli*, *S. aureus* and MRSA, as well as to promote proliferation of NIH 3T3 fibroblast cells [79]. Besides, Tet213-loaded Alg/HA/Col-dressing, applied one day after bacterial challenge and changed every 3 days for up to 14 days, induced wound healing, re-epithelialization, collagen deposition, and angiogenesis in an in vivo rat model of partial-thickness wounds with mixed-bacterial infection (*E. coli* and *S. aureus*) [79].

Numerous authors have been developing and applying delivery systems to enhance the properties of AMPs, which have proven to be encouraging approaches for treating non-healing infected DFUs. In fact, these systems protect AMPs from host diabetic microenvironment, protease degradation and serum inactivation, reduce their inherent toxicity and improve their targeting and prolonged delivery. Nonetheless, the use of delivery systems themselves can also interfere with wound healing, and consequently needs to be considered when assessing the efficacy of a given formulation. It is noteworthy that some delivery systems themselves also have antimicrobial properties and can moisten the wound microenvironment, which will facilitate the wound-healing process. Finally, these formulations will also prevent the emergence of bacterial or fungal resistance, therefore becoming attractive alternatives to the use of antibiotics and antifungals.

## 6. Conclusions and Future Perspectives

Despite a wealth of research about AMPs and their respective application as potential therapy for non-healing infected wounds, this area needs further investigation. There is evidence that the performance of chemical modifications and the use of delivery systems can greatly improve the characteristics of AMPs to be applied as alternatives to antibiotics and antifungals. AMP-based approaches could be a solution for the emergence of antimicrobial resistance or could be applied in association with antibiotics or antifungals to promote a synergistic action for treating chronic wounds. However, few have been developed to treat polymicrobial infections that include anaerobic bacteria, fungi, and biofilms, and consequently to improve the treatment of infected DFUs. Only Gomes et al., Tomioka et al., and Sultan et al. have evaluated the action of PP4-3.1, SR-0379 and K11R-K17R against fungi, respectively, without any study considering the action of AMPs against anaerobic bacteria present in the DFU microenvironment. Therefore, further studies will need to include more models of infection with anaerobic bacteria, fungi, and biofilms, since infected DFUs tend to have a multi-kingdom basis. Indeed, non-healing DFUs have been highly associated with fungal pathogens and anaerobic bacteria [11,12,13,14,17,18]. It is noteworthy that the infection models used in the different studies presented herein include microorganisms that are more pathogenic and predominant in DFUs, such as *S. aureus* (MSSA and MRSA), *P. aeruginosa*, *E. coli*, and *A. baumannii*, as well as some *Candida* spp. However, these infection models only include one or two of these microbes, and do not consider the complexity of polymicrobial infections and biofilms in human-infected chronic wounds. Furthermore, more accurate models of infected DFUs need to be included in future research to prove the efficacy of novel AMP delivery systems as therapeutic approaches for treating chronic infected wounds. Indeed, better wound models also need to be implemented to better mimic the human condition, including full-thickness infected wound models. Together, these future improvements could conduct to a greater translation into the clinical practice and consequently to a reduction of clinical trial failure rates, leading to effective management and treatment approaches for multi-kingdom infected DFUs, to enhance the health and the quality of life of these patients.

## Figures and Tables

**Figure 1 biomolecules-11-01894-f001:**
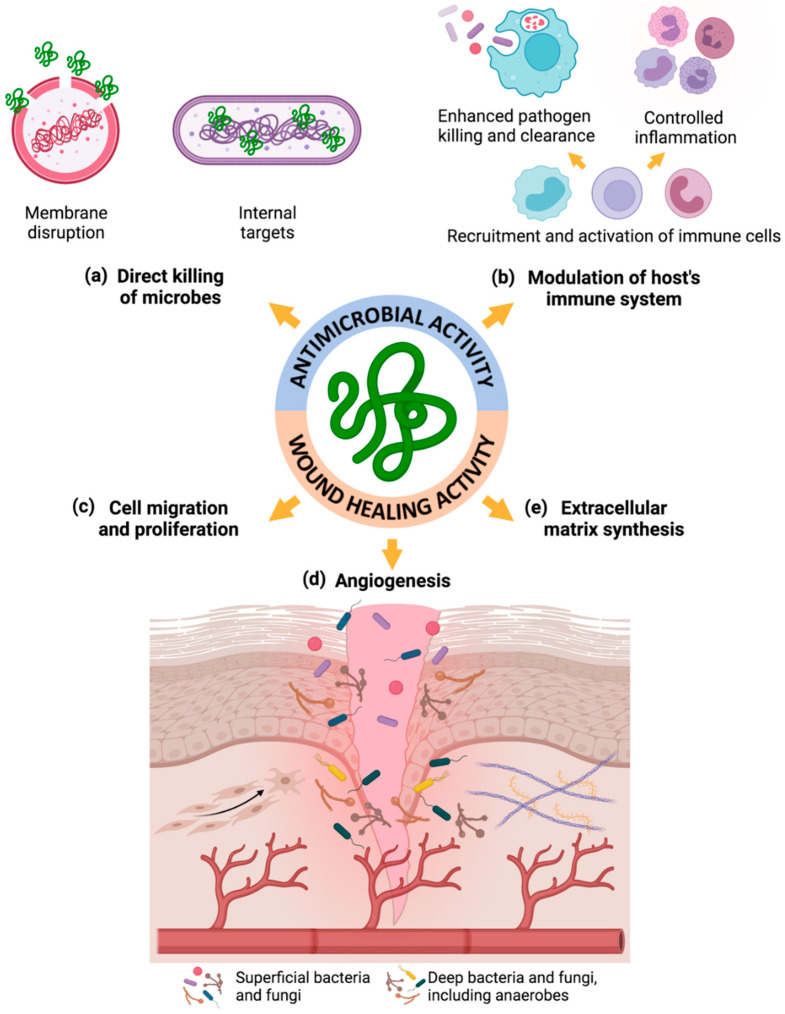
Mechanisms of action of AMPs supporting their therapeutic application for treating infected non-healing wounds—unraveled antimicrobial and wound-healing properties: (**a**) direct killing of microbes; (**b**) modulation of the host’s immune system; (**c**) promotion of cell migration and proliferation; (**d**) induction of angiogenesis; and (**e**) enhancement of extracellular matrix synthesis. Figure created in BioRender.com (accessed on 9 November 2021).

**Table 1 biomolecules-11-01894-t001:** Most predominantly identified microorganisms in DFUs, comprising both Gram-positive and Gram-negative bacteria, as well as anaerobic bacteria and fungi. All microorganisms are presented in order of the greatest abundance in DFUs.

Gram-Positive	BACTERIA Gram-Negative	Anaerobes
*S. aureus (MSSA* and *MRSA)* [8,9,10,14,15,17,32,34,35,36,37,38] *C. striatum* [10,32,34,43] *Streptococcus β-hemolytic* [8,9,17,18,32]	*P. aeruginosa* [8,9,10,14,17,35,36,37,38,44] *Proteus* spp. [8,9,17,37] *Enterobacter* spp. [8,17,37] *Citrobacter* spp. [8,17,42] *E. coli* [8,17,37] *A. baumannii* [8,17,38,43]	*Bacteroides* spp. [9,17,18] *Prevotella* spp. [9,10,17] *Peptostreptococcus* spp. [9] *Clostridium* spp. [9]
FUNGI		
*C. albicans* [11,12,13,14,19,20,39,40]	*C. tropicalis* [12,14,20,39,40]	*C. glabrata* [12,39,40]
*C. parapsilosis* [14,19,20,39,40]	*T. rubrum* [12,13,44]	*T. mentagrophytes* [12,13,40]
*A. fumigatus* [12,14,20]	*T. asahii* [14,19,20]	*C. herbarum* [19,20,40]

**Table 2 biomolecules-11-01894-t002:** Main endogenous AMP primary, secondary, and tertiary structures, and their related physicochemical properties, including length, molecular weight (MW), isoelectric point (pI), net charge, and hydrophobicity. PBD codes were obtained from the Protein Data Bank: www.rcsb.org (accessed on 1 December 2021). The physicochemical properties were obtained from www.pepdraw.com (accessed on 5 December 2021) and confirmed in other similar software, whereas the secondary and tertiary structures were obtained from www.compbio.dundee.ac.uk/jpred4/index.html (accessed on 7 December 2021) and www.rcsb.org/structure/ (accessed on 1 December 2021), respectively.

AMP	Primary Structure	Length (aa)	PDB Code	Secondary Structure	Tertiary Structure	MW (Da)	pI	Net Charge	Hydrophobicity (kcal/mol)
hBD-1	DHYNCVSSGGQCLYSACPIFTKIQGTCYRGKAKCCK	36	1IJU	α-helix + β-strand	three antiparallel β-sheets stabilized by three disulfide bridges and flanked by an α-helix segment, together stabilized by a disulfide bridge	3931.77	8.55	+4	+28.98
hBD-2	GIGDPVTCLKSGAICHPVFCPRRYKQIGTCGLPGTKCCKKP	41	1FD4			4331.17	9.26	+6	+32.25
hBD-3	GIINTLQKYYCRVRGGRCAVLSCLPKEEQIGKCSTRGRKCCRRKK	45	Not found			5157.70	10.47	+11	+45.26
LL-37	LLGDFFRKSKEKIGKEFKRIVQRIKDFLRNLVPRTES	37	2K6O	α-helix	one α-helical conformation	4490.57	11.15	+6	+41.03

**Table 3 biomolecules-11-01894-t003:** Free AMPs being applied as promising therapies for infected chronic wounds and their respective roles in antimicrobial and wound-healing activities. All AMPs are presented according to the following criteria: (1) free vs. chemically modified; (2) endogenous vs. synthetic; and (3) alphabetic order. AMP sequences are presented using the one-letter amino acid code, as per the IUPAC-IUBMB Joint Commission on Biochemical Nomenclature rules. ↑—increase; ↓—decrease. ^1^ AMPs that were tested against fungi.

AMP	Sequence	Source	Delivery Method	Role in Antimicrobial and Wound-Healing Activities	Reference
hBD-2 LL-37	GIGDPVTCLKSGAICHPV FCPRRYKQIGTCGLPGTK CCKKP LLGDFFRKSKEKIGKEFK RIVQRIKDFLRNLVPRTES	Endogenous (human) Endogenous (human)	Free Free	↑ antimicrobial activity (*E. coli*) ↑ keratinocyte migration	[23]
CW49	APFRMGICTTN	Synthetic (frog skin)	Free	↑ angiogenic ability ↑ anti-inflammatory effect little effect on re-epithelialization	[61]
IDR-1018	VRLIVAVRIWRR-NH2	Synthetic	Free	↓ in vitro toxicity compared to LL-37 ↑ wound healing in *S. aureus* infected porcine and non-diabetic but not in diabetic murine wounds	[62]
IDR-1018	VRLIVAVRIWRR-NH2	Synthetic	Free	↑ angiogenic ability ↑ anti-inflammatory effect ↑ migration of endothelial cells	[57]
Pexiganan	GIGKFLKKAKKFGKAFV KILKK	Synthetic (analogue of magainin II—frog skin)	Free	↑ antimicrobial activity (*E. coli*, *E. cloacae*, *Citrobacter* spp., *P. vulgaris*, *M. morganii*, *K. pneumoniae*, *S. marcescens*, *P. aeruginosa*, *A. baumannii*, *S. agalactiae*, *S. pyogenes*, *E. faecium*, *MSSA* and *MRSA*)	[58]
3.1-PP4	KKLLKWLLKLLKTTKS	Synthetic	Free (chemically modified)	↓ toxicity to HFF-1 human fibroblasts ↑ antimicrobial activity (*E. coli*, *P. aeruginosa*, and *K. pneumoniae*, including MDR isolates) ↓ formation of *K. pneumoniae* biofilms	[66]
PP4-3.1**^1^**	KTTKSKKLLKWLLKLL	Synthetic	Free (chemically modified)	↑ antimicrobial activity (Gram-positive and Gram-negative bacteria, including MDR isolates, as well as against relevant *Candida* spp.)	[67]
A-hBD-2	APKAMVTCLKSGAICHP VFCPRRYKQIGTCGLPGT KCCKKP	Synthetic	Free (chemically modified)	↑ structural stability ↓ toxicity to keratinocytes ↑ antimicrobial activity (*S. aureus*) ↑ migration and proliferation of keratinocytes ↓ terminal differentiation of keratinocytes ↑ mobilization of intracellular Ca^2+^ ↑ wound healing in vivo	[26]
LFcinB	FKCRRWQWRMKKLGAP SITCVRRAF	Synthetic (derived from bLF)	Free (chemically modified)	↑ keratinocyte migration in vitro and ex vivo ↑ wound healing ↑ antimicrobial activity (*B. pumilus* and *S. aureus*) ↑ angiogenesis and collagen deposition ↓ inflammation	[68]
SHAP1	APKAMKLLKKLLKLQK KGI	Synthetic	Free (chemically modified)	↓ toxicity to human erythrocytes and keratinocytes ↑ stability to proteases exposure ↑ wound closure compared to LL- 37 in vitro ↑ healing in vivo full-thickness excisional wounds ↑ antimicrobial activity (*S. aureus*) ↑ healing in S. aureus-infected murine wounds	[69]
SR-0379**^1^**	MLKLIFLHRLKRMRKRL DLysRK	Synthetic	Free (chemically modified)	↑ proliferation of human dermal fibroblasts ↑ antimicrobial activity (bacteria, including drug-resistant, and also fungi, namely: *E. coli*, *P. aeruginosa*, *S. aureus*, *C. krusei*, *T. mentagrophytes*, *T. rubrum*, *MRSA* and *A. baumannii* (MDR)) ↑ accelerated wound healing in two different wound-healing rat models	[47]

**Table 4 biomolecules-11-01894-t004:** AMPs loaded on delivery systems being applied as promising therapies for infected chronic wounds and their respective roles in antimicrobial and wound-healing activities. All AMPs are presented according to the following criteria: (1) loaded on a delivery system vs. chemically modified and loaded on a delivery system; (2) endogenous vs. synthetic; and (3) alphabetic order. AMP sequences are presented using the one-letter amino acid code, as per the IUPAC-IUBMB Joint Commission on Biochemical Nomenclature rules. ↑—increase; ↓—decrease. ^1^ AMPs that were tested against fungi; ^2^ AMPs that were/are under clinical trials.

AMP	Sequence	Source	Delivery Method	Role in Antimicrobial and Wound-Healing Activities	Reference
hBD-1 HNP-1	GNFLTGLGHRSDHYNCV SSGGQCLYSACPIFTKIQ GTCYRGKAKCCK EPLQARADEVAAAPEQIAADIPEVVVSLAWDESL APKHPGSRKNMACYCRI PACIAGERRYGTCIYQGR LWAFCC	Endogenous (human) Endogenous (human)	Niosomal gel Niosomal gel	↑ antimicrobial activity (*MRSA*-infected wound in rats and *MSSA* and *MRSA* isolated from patients with DFIs)	[70]
Nisin	ITSISLCTPGCKTGALMGCNMKTATCH(or N)CSIHVSK	Endogenous (bacteria)	Guar gum gel	↑ antimicrobial activity against *S. aureus* DFU biofilm-producing isolates, including some multidrug-resistant clinical isolates	[25]
Nisin	ITSISLCTPGCKTGALMGCNMKTATCH(or N)CSIHVSK	Endogenous (bacteria)	Guar gum gel	↑ antibacterial activity against biofilms formed by DFI *S. aureus*	[71]
aCT1 ^2^	RQPKIWFPNRRKPWKKRPRPDDLEI-acid	Synthetic (analogue of Cx43)	Hydroxyethyl cellulose gel	↓ ulcer area in DFU patients ↑ ulcer re-epithelialization in DFU patients ↓ time-to-complete-ulcer closure in DFU patients	[56]
ASP-1 ASP-2	RRWVRRVRRWVRRVVRVVRRWVRR RWWRWWRRWWRR	Synthetic	Gel, Stratex or PU-based dressings	↑ eradication of mono- and polymicrobial biofilms of MDR pathogens: *S. aureus*, *A. baumannii*, *K. pneumoniae*, *P. aeruginosa*, and *MRSA* ↑ BI compared to free ASP-1 and ASP-2	[72]
IKYLSVN	IKYLSVN	Synthetic	GOx-loaded hydrogel	↑ antimicrobial activity (*S. aureus*) ↓ blood glucose concentration of diabetic patients	[73]
LL-37	LLGDFFRKSKEKIGKEFK RIVQRIKDFLRNLVPRTE SC	Synthetic	Gold-nanoscale formulation	↑ phosphorylation of EGFR and ERK1/2 ↑ migratory properties of keratinocytes ↑ wound-healing activity in vivo ↑ expression of collagen, IL6 and VEGF	[48]
Pexiganan ^2^	GIGKFLKKAKKFGKAFV KILKK	Synthetic (analogue of magainin II—frog skin)	Cream	=clinical outcome, microbiological eradication (*S. aureus*, *E. coli*, *E. cloacae*, *S. marcescens*, *P. aeruginosa*, *Enterococcus* spp., *MSSA* and *MRSA*), and wound healing as ofloxacin ↓ bacterial resistance in vivo	[74]
Cys-KR12	CKRIVKRIKKWLR	Synthetic (originated from LL37)	SF nanofiber membrane (chemically modified)	↑ antimicrobial activity (*S. aureus*, *S. epidermidis*, *E. coli*, and *P. aeruginosa*) ↑ proliferation of keratinocytes and fibroblasts ↑ differentiation of keratinocytes ↓ LPS-induced TNF-α expression of monocytes	[75]
K11R-K17R**^1^**	DSHAKRHHGYRRKFHERHHSHRGY	Synthetic (analogue of Hst-5 peptide)	HPMC-based bioadhesive hydrogel (chemically modified)	↑ antimicrobial activity (*C. albicans* strains resistant to traditional antifungals) ↑ cell proliferation and migration in human oral keratinocytes	[76]
KSL-W	KKVVFWVKFK	Synthetic (analogue of KSL peptide)	Pluronic F-127 gel (chemically modified)	↑ antibiofilm and antimicrobial activity (chronic wound infection biofilm-embedded bacteria, including *MRSA*, *S. epidermidis*, *CoNS*, and *A. baumannii*)	[77]
TC19	LRCMCIKWWSGKHPK	Synthetic (derived from human TC-1-derived peptide L3)	HPMC gel (chemically modified)	↓ toxicity to human fibroblasts ↑ antimicrobial activity (ESKAPE panel in vitro, and *MRSA* and *A. baumannii* (MDR) in a murine superficial wound infection model) ↓ bacterial resistance inflammation in vitro	[78]
Tet213	KRWWKWWRRC	Synthetic (cysteinylated HHC36 peptide)	Alg/HA/Col dressing (chemically modified)	↑ antimicrobial activity (*E. coli*, *S. aureus*, *MRSA*) ↑ proliferation of NIH 3T3 fibroblast cells ↑ wound healing, re-epithelialization, collagen deposition, and angiogenesis in vivo rat model of partial-thickness mixed-bacterial infected wounds	[79]

## Data Availability

Not applicable.

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
