# Peer review of "Bioactive Antimicrobial Peptides as Therapeutic Agents for Infected Diabetic Foot Ulcers"

_biomolecules, 2021, doi:10.3390/biom11121894_

Round 1

Reviewer 1 Report

Diabetic foot ulcer is a complex and chronic wound in patients with diabetes mellitus that requires a multidisciplinary approach for its control and treatment, mainly due to resistance to antibiotics. Peptide-based therapies point to translational avenues for managing this serious complication. In recent years, antimicrobial peptides have been the main focus of numerous publications and bibliographic reviews as a result of their importance in the current pharmaceutical market. However, its application to this harmful problem of diabetes mellitus is underrecognized. For this reason, this manuscript presents useful and up-to-date information that reinforces the immense potential of the peptide and broadens the vision of its applicability, with an emphasis on the diabetic foot ulcer. The study is relevant. The suggestions presented here should strengthen this work.

  1. The primary structure of the peptides with promising activity was shown in table 2. However, little information was presented and discussed about the physicochemical properties of these peptides. Authors are invited to use bioinformatics tools in order to calculate the main properties and discuss them.
  2. What is the secondary or tertiary structure of these peptides?
  3. It would be appropriate to discuss the main models of infected DFU. What are the advantages and disadvantages?
  4. The pros and cons of delivery methods should also be discussed.
  5. Are there investigations that have evaluated the synergism of peptides and other drugs in models of infected DFUs? Which drugs/antibiotics have already been tested or which would be interesting to be evaluated?
  6. What are the main chemical modifications performed on peptides with the potential for infected chronic wounds? This information should be included.
  7. The potency and concentration ranges that the peptides exhibit activity were not shown. Considering the clinical translation, this information is valuable, as well as for future comparative studies.
  8. The target therapeutic effects were well described, however it would be relevant to detail the cytotoxicity of the peptides presented . If they were previously evaluated in healthy cells or red cells, it would be appropriate to present the IC50 values.
  9. Gram should be capitalized when used as Gram stain.

Reviewer 2 Report

The submitted manuscript “Bioactive antimicrobial peptides as therapeutic agents for infected diabetic foot ulcers” by Jessica et al highlighted all the major functions and applications of antimicrobial peptides (AMPs) for infected diabetic foot ulcers. This is a great topic and lacks a summary of recent years' study on this topic. They have summarised and organised very well and easy for the readers to follow. I highly recommend this manuscript be published after minor revision.

There are a few minor comments,

  1. Some abbreviations, for example, antimicrobial peptides (AMPs) was frequently used in the text but not consistent.

Also, line 266, shall “AMPS” be “AMPs”? This typo has appeared a couple of times.

  1. Section 3 on page 4, the authors introduced AMPs, there are multiple excellent reviews on AMPs published should be referred (Lancet Infect Dis 2020; 20: e216–30, https://doi.org/10.1016/S1473-3099(20)30327-3, Chem. Soc. Rev., 2021,50, 4932-4973 https://doi.org/10.1039/D0CS01026J).
  2. Line 456, this sentence here doesn’t make sense.
  3. It will be better to cite Table 2 carefully in the text of sections 5.1 and 5.2, which will guide the reader to follow and track.

Reviewer 3 Report

I really appreciate authors efforts to provides a very important and timely focus review article . The review article demonstrated an extensive focus on  bioactive antimicrobial peptides as therapeutic agents for infected diabetic foot ulcers.

The article is well planned, reasonably written and  well discussed. 

It was very interesting while going through the manuscript. However, I suggest authors to include attest one more graphical representation or schematic diagram to attract high attention with the readers of the MDPI-Biomolecules.

Round 2

Reviewer 1 Report

The authors have addressed all of my comments. It is an interesting and well-written manuscript, which contribute significantly to the field. In this context, I have no further concerns regarding the manuscript.